# Study protocol to examine the effects of acute exercise on motor learning and brain activity in children with developmental coordination disorder (ExLe-Brain-DCD)

**Albert Busquets**[1]*, **Blai Ferrer-Uris**[1], **Turgut Durduran**[2,3], **Faruk Bešlija**[2], **Manuel Añón-Hidalgo**[1], **Rosa Angulo-Barroso**[1,4]*

**1** Institut Nacional d'Educació Física de Catalunya, University of Barcelona, Barcelona, Spain, **2** Institut de Ciències Fotòniques, The Barcelona Institute of Science and Technology, Castelldefels, Spain, **3** Institució Catalana de Recerca i Estudis Avançats, Barcelona, Spain, **4** Kinesiology, California State University, Northridge, California, United States of America

* albert.busquets@gencat.cat (AB); rosa.angulobarroso@csun.edu (RAB)

## Abstract

### Introduction

Developmental coordination disorder (DCD) is one of the most prevalent pediatric chronic conditions. Without proper intervention, significant delays in motor skill performance and learning may persist until adulthood. Moderate-to-vigorous physical exercise has been proven to improve motor learning (adaptation and consolidation) in children with or without disorders. However, the effect of a short bout of physical exercise on motor adaptation and consolidation in children with DCD has not been examined. Furthermore, the role of perceptual-motor integration and attention as mediators of learning has not been examined via neuroimaging in this population.

### Objectives

Therefore, the primary aims of this project will be to compare children with and without DCD to (a) examine the effect of acute exercise on motor learning (adaptation and consolidation) while performing a rotational visuo-motor adaptation task (rVMA), and (b) explore cortical activation in the dorsolateral- and ventrolateral-prefrontal cortex areas while learning the rVMA task under rest or post-exercise conditions.

### Methods

One hundred twenty children will be recruited (60 DCD, 60 controls) and within-cohort randomly assigned to either exercise (13-minute shuttle run task) or rest prior to performing the rVMA task. Adaptation and consolidation will be evaluated via two error variables and three retention tests (1h, 24h and 7 days post adaptation). Cortical activation will be registered via functional near-infrared spectroscopy (fNIRS) during the baseline, adaptation, and consolidation.

**Data Availability Statement:** After finishing the study and with the express authorization of the

Data Protection Officer of the Department of Education of the Generalitat de Catalunya, the data processed and anonymized will be available in B2share repositories (https://b2share.eudat.eu/) under the Creative Commons CC BY-NC-SA license.

**Funding:** AB, BF, FB, and RA as authors of this study that is part of the R+D+i project PID2020-120453RB-I00 received funding from the Ministerio de Ciencia e Innovación – Agencia Estatal de Invenstigación (https://www.aei.gob.es/; MCIN/AEI/10.13039/501100011033/). In addition, MA earned a PhD fellowship funded by the Institut Nacional d'Educació Física de Catalunya (INEFC) of the Generalitat de Catalunya (https://inefc.gencat.cat/es/inefc_barcelona/). The funders had and will not have a role in the study design, data collection and analyses, decision to publish, or preparation of the manuscript.

**Competing interests:** AB; BF, FB, and RA as authors of this study that is part of the R+D+i project PID2020-120453RB-I00 received funding from the Ministerio de Ciencia e Innovación – Agencia Estatal de Investigación (https://www.aei.gob.es/;MCIN/AEI/10.13039/501100011033/). In addition, MA earned a fellowship funded by the Institut Nacional d'Educació Física de Catalunya (INEFC) of the Generalitat de Catalunya(https://inefc.gencat.cat/es/inefc_barcelona/). The funders had and will not have a role in the study design, data collection and analyses, decision to publish, or preparation of the manuscript.

## Discussion

We expect to find exercise benefits on motor learning and attention so that children with DCD profiles will be closer to those of children with typical development. The results of this project will provide further evidence to: (a) better characterize children with DCD for the design of educational materials, and (b) establish acute exercise as a potential intervention to improve motor learning and attention.

## Introduction

### Developmental coordination disorder: Functional consequences

World Health Organization [1] has defined developmental coordination disorder (DCD) as a developmental disorder characterized by a significant delay in the acquisition of gross and fine motor skills and impairment in the execution of coordinated motor skills that manifests in clumsiness, slowness, or inaccuracy of motor performance. Coordinated motor skills are substantially below those expected given the individual's chronological age and level of intellectual functioning [2].

These motor coordination problems make daily activities and participation in physical activities difficult and unappealing. As a consequence, a cascading effect on academic [3], physical [4–9] and emotion health [3, 10–14] issues takes place, and can have a great impact in the quality of life [15–17] of these individuals.

### Developmental coordination disorder: Comorbidity and severity

Although DCD is a unique and separate neurodevelopmental disorder, its manifestations are multiple and diverse producing large heterogeneity among those who receive the diagnosis [18]. It is known that about 35–50% of children with DCD also have attention deficit hyperactivity disorder (ADHD) [3, 19, 20], and 50% of dyslexic children also have DCD [21]. However, for the purpose of the present line of research, children with probable DCD, with or without comorbid ADHD or dyslexia, will be in focus. This decision will allow us to characterize and study this population as it presents itself in most contexts.

The diagnosis of DCD cannot be explained by mental retardation, specific congenital or acquired neurological disorder [22]. On the contrary, DCD should be determined by poor motor coordination that interferer with academic performance or activities of the daily life. The Motor Assessment Battery for Children (M-ABC2, [23, 24]) and specific check-lists and questionnaires are used to determine DCD severity.

Because DCD is a chronic condition that cannot be reversed and will continue until adulthood [25], preventive and intervention actions are crucial [26]. In fact, many have proposed that a child with a diagnosis of DCD should be treated and that any treatment is better than no treatment [22, 27, 28]. Unfortunately, even when diagnosed, DCD often goes untreated [29]. However, in the last decade, studies examining the effect of different forms of chronic exercise on motor and cognitive performance of children with DCD have proliferated (for reviews, see [30–34]).

### Motor learning and DCD

Problems involved in the acquisition (that is, learning), not only the execution, of coordinated motor skills are now a critical characteristic in the identification of children with DCD. Surprisingly, little research is available focusing on how children with DCD learn new motor skills [35].

Recently, a plausible hypothesis has been put forward to explain the impaired motor learning in children with DCD. This hypothesis suggests a deficit in the generation of an internal model resulting in children with DCD having a reduced ability to use predictive (feedforward) motor control [32, 36–39].

Successful motor control is thought to result from an internal model that predicts sensory consequences of a motor command. DCD children may have difficulties creating adequate internal models because of alterations in their fronto-parietal circuits, which are linked to anticipatory planning and online monitoring of movement [36, 40]. Therefore, cerebellar network connections to frontal and parietal areas also seem to be implicated in DCD, providing indirect support for the internal modeling hypothesis [41–44].

The internal model generation capacity in children with DCD has been studied using rotational visuomotor adaptation (rVMA) tasks. Kagerer et al. [45] exposed that DCD children presented a weak modification of their internal model during the adaptation phase of the rVMA task, and suggested that these children already had a poorly defined visuomotor internal model in the baseline condition (i.e., larger variability). A follow-up study conducted by Kagerer et al. [46] indicated that DCD children presented poorer adaptation than TD children with in a similar number of task trials. Moreover, they observed that when the sensory perturbation was gradually presented (i.e., increments of 10˚ of perturbation to achieve 60˚), children with DCD did not appear to be able to use the small differences in perceptive signals to adapt their internal model. These results seem to indicate that: (1) DCD children may be able to adapt their internal model but they will need enough sensory discrimination to generate error feedback (i.e., the alteration must be larger than variability); and (2) the learning rate of DCD children may be slower than TD children [19].

Surprisingly, and despite the prefrontal cortex's (PFC) critical role in the internal model adaptation mediating contingencies of action because of the interplay of sensory and motor working memories [47, 48], little attention has been placed on this brain area in motor adaptation studies involving children with DCD. Activation of the PFC is expected during situations where the relationship between visual and proprioceptive information is modified (i.e., motor adaptations tasks). Concretely, the dorsolateral prefrontal cortex (DLPFC) is activated during the early stages of a new motor behavior, due to its engagement in processing sensory inputs and planning future actions, but its activation declines as learning is acquired [49, 50]. On the other hand, the ventrolateral prefrontal cortex (VLPFC) is engaged during the learning of a new situation but also when motor adaptation is acquired, probably because of its contribution to the maintenance of the spatial information and to the suppression of motor responses to an irrelevant stimulus (i.e., inhibition) [50, 51]. Therefore, studies using neuroimaging of the prefrontal areas during the motor adaptation task could increase our knowledge about the underlying mechanisms of the DCD in children and provide new evidence that could support theoretical approaches, such as the internal modeling hypothesis.

## Acute exercise and motor learning

Exercise, and particularly, acute exercise could be an enhancer of function [52–54], including motor learning [55, 56]. However, evidence for an acute exercise-motor learning relationship does not exist for children with DCD. Furthermore, very few studies have examined this question in children with TD. To our knowledge, only three studies explored the effect of acute exercise on learning a motor task in children with TD. In Lundbye-Jensen et al. [57] study, the consolidation of a tracking task was improved when participants performed an acute bout of exercise. Ferrer-Uris and collaborators [55] analyzed the effects of acute exercise on the acquisition and consolidation of an rVMA task in TD children. They observed how a 13-minute

acute bout of exercise enhanced consolidation, especially when exercise was performed before practicing the motor task [55]. Furthermore, this research group also observed how even a shorter exercise bout of only 5 minutes could enhance motor learning (consolidation) [56]. Therefore, despite the scarce information regarding the acute exercise effects on children's motor learning, the existing evidence points out that exercise, even when it is as short as 5 minutes, could potentially enhance the learning in TD children. Since children with DCD have motor learning deficiencies, we set up this project to address whether they could benefit from an acute exercise intervention.

Despite the limited knowledge about why exercise contributes to learning enhancements, the mechanisms underlying these benefits have been related to changes in brain activity and concentration increases in neurochemicals [58]. Studies utilizing functional near infrared spectroscopy (fNIRS) have shown brain activation changes due to exercise in adult and elderly [59, 60]. Therefore, concurrent brain imaging and behavioral assessments would be the appropriate next step to explore the exercise benefits on learning in children. To our knowledge, there are no such studies, and certainly not in children with DCD.

## Attention as a potential mechanism to explain deficits in motor learning in DCD

Children with DCD have more attentional problems than their peers [3]. Specifically, Fong et al. [61] found that children with DCD were less attentive to M-ABC movements than their peers, even when subjects with comorbid ADHD were eliminated. These authors used a one-channel electroencephalography (EEG) placed over the PFC to assess attention [62], showing that the attention index was significantly associated with M-ABC motor impairment so poor motor performance was explained by inattention after controlling for age, sex, body mass index, and physical activity level.

In this regard, DLPFC has been linked, together with anterior cingulate, to attentional control [63, 64] while VLPFC has been found to be engaged in visuo-attentional tasks, due to its link with the inferotemporal cortex [65–67]. Using fMRI previous research has shown how DCD children present less prefrontal activity compared to TD children in various attention-dependent cognitive and motor tasks, especially in the DLPFC area [41, 44, 68]. These studies suggest that poorer attentional capability may be one factor that impacts motor skill acquisition in children with DCD and neuroimaging can help elucidate these differences. However, whether these differences are also mediated by the visuomotor integration network in children with DCD has not been examined.

Although EEG and fMRI are widely used techniques, optical technologies provide less invasive and more practical way to examine brain activation in pediatric populations [69–74]. fNIRS technique uses light to measure changes in cerebral oxygenation and deoxygenation fractions of hemoglobin ($[O_2Hb]$ and $[HHb]$, respectively). It has been used in adults to reliably determine different levels of brain activity in different conditions, including the activity of regions related to sustained attention [75]. Byun et al. [76] used this technique to find evidence about how acute exercise improved executive function via the exercise-induced arousal system. Similarly, fNIRS has been used in pediatric populations with and without disabilities to examine attention and executive function (response inhibition, cognitive shifting, working memory, and attention, for a review see, [77]). Reliability of fNIRS as the measurement of brain activation in children has been proven [78]. Recently, Caçola et al. [79] used fNIRS to examine cortical activation in children with DCD while performing different tasks (finger tapping, curve tracing, and paragraph writing). DCD children compared to TD children showed different focal activation patterns, even for the finger tapping (a simple task with no spatial or temporal

demands), and these differences were also task-specific (for example, differences in the right DLPFC were found for the paragraph writing).

To our knowledge, no studies have utilized optical neuroimaging in children with DCD to explore brain responses in a motor learning context, much less the impact of acute exercise, and this project is a first attempt to provide information in these cases.

## Attention: Exercise as an enhancer of learning via attention

Although the specific underlying mechanisms by which physical exercise can improve learning and memory are not clear, several proposals have been placed forward. From the psycho-cognitive approach, it is proposed that arousal and attention may mediate the relationship between physical exercise and improved learning. An optimal level of arousal is necessary to effectively use the limited attention capacity [80]. This effective use of attention, in turn, facilitates learning [81].

Therefore, it is plausible that moderate acute physical exercise places the individual at the appropriate level of arousal (not too high, not too low) and assists in the maintenance of attention, so that the learning and the consolidation of a new motor-perceptual task, like the rVMA task, are improved. In fact, Medina et al. [82] found that the attention deficits in children with ADHD could be minimized via physical exercise using an acute interval exercise of 30 minutes and assessing sustained attention using the Conner's Continuous Performance II test. However, these authors had no measurements of learning, memory or neuroimaging.

## Justification

This project is designed to provide a multidisciplinary link between the educational sciences, sports sciences, and neurosciences to better understand the relevant population of children suffering from DCD. Results from this project have translational applications to teachers, parents, and other children educators that will allow elaborating adequate interventions from solid evidence (i.e., science-based interventions). The key concept is the promising idea that physical exercise is the "new preventive medicine" not only for chronic vascular and metabolic diseases, but also a potential enhancer of learning, even when it is resulting from an acute bout of exercise. Children are at a developmental phase where the brain is still very plastic and establishment of enduring healthy habits is more effective. Children with DCD have problems with learning motor skills, which are the support for a long-lasting active life style and becoming physically active adults. Unfortunately, therapeutic interventions are not the main stream for this population despite its high prevalence (6%) [83].

Motor learning entails integration of motor and sensory information and is constrained by our capacity to sustain attention. It is suggested that difficulties to perform motor tasks by children with DCD are related to poor visuomotor integration and attention deficits. Our previous work suggested that the improvements in learning of a motor skill are related to the experience in exercise and sport [55, 84, 85] even when the learners suffer some disorder or functional incapacity [86–88]. Furthermore, previous studies [55, 56, 89] have demonstrated that the acute intense exercise enhances the motor learning in adults and children with typical development. However, whether the children with DCD could benefit from this strategy to improve their motor learning capacity is not known. The scarce brain studies conducted with adults about the exercise-learning interaction [90] indicate that the acute exercise benefits on motor learning may be associated, in part, to the changes in the brain activity. However, no similar studies in children, much less in DCD children, have been published so far. Therefore, the concurrent brain imaging and behavioral assessments would be the next logical step to explore the exercise benefits on learning in children with or without DCD.

In order to examine the combined effect of the acute exercise and learning on children's brain activity, widely used neuroimaging techniques are either too expensive, or do not allow for in-situ assessment (i.e., same location where learning is occurring, in our case, children's schools). Optical technologies have been utilized and suggested as potential and promising tools to study the alterations of the brain networks. Perhaps most importantly, the features of this technological approach (non-invasive, quantitative, safe, inexpensive and portable) and the characterization of brain activity using fNIRS will allow us to conduct studies in-situ, with robust results, including estimation of which brain areas are activated during the learning in children with DCD, and to which extent. All this knowledge is hypothesized to assist in the design of better, more targeted interventions. In addition, exercise benefits in learning, if confirmed in this population, could also be integrated as a part of the effective educational interventions for children with DCD.

# Materials and methods

## Objectives

Although the long-term objective of this line of research will address the underlying mechanisms of acute exercise effects on motor learning and possible brain activity changes with developmental coordination disorder (DCD), we currently defined the specific objectives of this project as follows:

- O1. To study acute intense exercise effects on motor learning (adaptation and consolidation) in children with and without DCD from a behavioral perspective (performance and execution).

- O1.1- Characterize possible motor learning modulators (physical fitness; physical activity level; attentional and cognitive capabilities) in children with and without DCD.

  ○ O1.2- Compare motor learning (adaptation and consolidation) ability in children with and without DCD while performing a rotational visuo-motor task (rVMA).

  ○ O1.3- Examine the effect of acute intense exercise on motor learning (adaptation and consolidation) ability in children with and without DCD while performing a rotational visuo-motor task.

- O2. To explore brain activity in dorsolateral pre-frontal cortex (DLPFC) and ventrolateral pre-frontal cortex (VLPFC) while learning a motor adaptation task in children with and without DCD, under rest or post-exercise conditions.

  ○ O2.1- Assess changes in DLPFC and VLPFC activity ($[0_2Hb]$) during rVMA learning (adaptation and consolidation) in typically developing children and in children with DCD.

  ○ O2.2- Evaluate the impact of acute intense exercise on changes in DLPFC and VLPFC activity ($[0_2Hb]$) during rVMA learning (adaptation and consolidation) in typically developing children and in children with DCD.

  ○ O2.3- Associate the presence of changes in the cortical activation variables with differences shown in behavioral variables.

## Study design and setting

The proposed study will be a four-group randomized control trial (RCT) with repeated measures where children with DCD and TD will be assigned randomly to control groups

(CON-DCD, CON-TD) and exercise-based intervention groups (EX-DCD, EX-TD) achieving a 1:1:1:1 allocation ratio within each school. The TD children cohort and its subsequent groups will be matched to the DCD cohort and its subgroups using age, sex at birth, and, handedness because these factors could affect the acute exercise effect on cognitive performance [91], as well as the development of visuomotor representations [92]. The intervention nature (i.e., performing exercise before the learning task) will not allow us to blind the participants or researchers; however, researchers in charge of the data reduction and analysis procedures will be blinded thanks to the data bases pseudonymization. The study will take place in different schools of Barcelona, Spain and the recruitment of the participants is scheduled to begin in October 2, 2023, and it will extend maximum to March 28, 2025. The study protocol was developed in accordance with the Standard Protocol Items Recommendations for Interventional Trials (SPIRIT) (S1 Checklist). The protocol has been registered at ClinicalTrials.gov (ID: NCT05936372).

## Participants

A total of 120 participants between 7.5 and 10.5 years, attending 2nd, 3rd or 4th primary school years, and divided into two cohorts (60 children with DCD and 60 TD children) will be recruited for the study. Taking into consideration the prevalence of 5–6% for DCD in the children population [22, 38, 93–96], screening a total of 1000 children using a motor-based questionnaire completed by parents and by teachers (M-ABC2 checklist, [24]) will be the first step to identify the children with DCD on the basis of their difficulties in daily living activities and academic performance. The children identified as potentially having a DCD by the motor-based questionnaire will be then evaluated to confirm their inclusion in the DCD group. Children will be selected into the DCD group based on the following inclusion criteria: (1) a movement assessment battery for children–second edition (MABC-2, [24]) score of <15% administered by trained testers. The MABC-2 is a standardized and normative referenced test designed to identify motor impairment in children aged 3–16 years by evaluating manual dexterity, ball skills, and static and dynamic balance. Because DCD children present motor performance substantially below that expected given the person's chronological age and intelligence, a <15% cut-off is usually used to indicate children with probable DCD [22, 97]. The Spanish translated edition and norm-referenced in Spain MABC-2 [98] will be used for scoring. (2) An average or better cognitive ability tested through the Test of Nonverbal Intelligence version 4 (TONI-4; [99]) and scoring above 85 in the intellectual quotient (IQ). (3) A parent-report history to confirm that motor difficulties showed by their child cannot be explained by any other neurological, developmental, and/or severe psychosocial problem according to the child's pediatrician. Comorbid attention deficit hyperactivity disorder, attention deficit disorder, and dyslexia will be acceptable in order to better represent the DCD population since data population-based studies suggest that almost 40% of the children with DCD have combined problems related to learning and/or attentional disorders [22, 96].

The TD cohort will demonstrate usual levels, given their age, performing daily activities and academic performance, with a 25th percentile or greater score on M-ABC2 [100], an average cognitive ability (>85 IQ in the TONI-4; [99]), and a parent-reported history confirming that their child does not have any problem and/or disability affecting motor learning according to the child's pediatrician. Uncorrected 20/20 vision will be the exclusion criterion for all children.

Participants of both cohorts (DCD and TD) will be assigned using partial–within cohort randomization to one of the two groups depending on whether they are participating in an exercise bout prior to learning the task ($t_0$ in Fig 1): (1) EX, who will engage in the learning

| Instruments and assessments | | Study period | | | | | |
|---|---|---|---|---|---|---|---|
| | | Enrollmen | Session 1 | Allocation | Session 2 | Session 3 | Session 4 |
| **Timepoints** | | $t_{-2}$ | $t_{-1}$ | $t_0$ | $t_1$ | $t_2$ | $t_3$ |
| **ENROLMENT** | | | | | | | |
| Eligibility screen | PARQ | X | | | | | |
| | Parent-reported neurological and developmental status screen | X | | | | | |
| | M-ABC2 checklist | X | | | | | |
| | M-ABC2 | | X | | | | |
| | TONI-4 | | X | | | | |
| Informed consent and participant assent | | X | | | | | |
| **ALLOCATION** | | | | X | | | |
| **INTERVENTIONS** | | | | | | | |
| Intense endurance exercise (iEE) | | | | | X | | |
| Rest (control intervention) | | | | | X | | |
| **ASSESSMENTS** | | | | | | | |
| Participant characterization | PAQ-C | X | | | | | |
| | Comorbidities to DCD | X | | | | | |
| | Conners CBRS | X | | | | | |
| | Anthropometry | | X | | | | |
| | Physical level (20mSRT) | | X | | | | |
| Family characterization | Family history about DCD and neurological disorders | X | | | | | |
| | Family's socio-economic status | X | | | | | |
| | Parental education | X | | | | | |
| Rotational Visuomotor Adaptation task (rVMA) | | | | | X | X | X |
| Cerebral activity (fNIRS) | | | | | X | X | X |
| Heart rate | | | X | | X | | |
| Pulseoxymetry | | | | | X | X | X |

**Fig 1. SPIRIT schedule of study's enrolment, interventions, and assessments.**

task after exercising, and (2) CON, who will rest until the learning task. Participants will not have prior experience with the proposed learning task (i.e., the rVMA).

## Sample size calculation

Statistical power analyses were performed for the sample size estimation, based on initial directional error (IDE, measured in degrees) data from Kagerer et al. [46] study (N = 20) comparing TD (3.96 ± 2.12) to DCD (1.62 ± 1.56) and from Ferrer-Uris et al. [55] study (N = 21) comparing EX (17.39 ± 4.03) and CON (22.95 ± 8.49) groups. The effect sizes (ES) in these studies were d = 1.26 and d = 0.8 (considered to be large, using Cohen's [101] criteria). With alpha = .05 and power = 0.80, the projected sample size for an inter-group comparison (TD vs. DCD or CON vs EX) needed to achieve these effect sizes is approximately between N = 12 and N = 26 per group calculated using GPower 3.1 or N = 18 per group following Cohen's [101, 102] criteria. Thus, our proposed sample size of 120 children (60 TD and 60 DCD; 30 EX and

30 CON within each cohort) should be adequate for the main objectives of this study and should also allow for expected attrition and our additional objectives of controlling for possible moderating factors, subgroup analysis, etc. Controlling the covariates will reduce residual error and further increase power.

## Interventions

Participants will engage in four different sessions (see Fig 1). Before enrolling in any of the study sessions ($t_{-2}$), participants, along with their guardians, will complete a set of questionnaires to assess: (1) the participant's health status to participate in physical activity (Physical Activity Readiness Questionnaire, PAR-Q); (2) the participant's physical activity engagement (Spanish version of the Physical Activity Questionnaire for Children, PAQ-C, validated by Benítez-Porres et al. [103]); (3) the participant's academic performance and difficulties performing daily living activities (Spanish version of the M-ABC2 checklist, validated by Ruiz & Graupera-Sanz [98]); (4) the possible comorbidities permitted in this study, such as attention deficit hyperactivity disorder, attention deficit disorder, and/or dyslexia; (5) the type and the dose of the medication a child currently takes; (6) possible psychosocial problems assessed by the Conners Comprehensive Behavior Rating Scales (Conners CBRS; [104]); (7) the exclusion criteria via checklist; and (8) the family history about DCD, neurological disorders or mental health, and family's socio-economic status (SES) and parental education. At the same time, participant's teacher will also complete the M-ABC2 checklist [98] questionnaire to identify the motor coordination problems and the participant's academic level. Participant handedness will be determined based on their preferred hand for everyday activities and confirmed by the M-ABC criteria.

In the first session ($t_{-1}$), participants' intelligence level will be assessed through TONI-4 and their motor development will be evaluated by MABC-2. Basic anthropometric parameters (height and body mass) will be also obtained and used to calculate body mass index (BMI). The fitness level (estimated $VO_2$max) will be assessed through the 20-meter shuttle run test (20mSRT) [105]. To further support children, an adult will run next to the child to assist in keeping the pace and to provide encouragement. The last completed stage (Emax) of the test will be recorded. In addition, during the 20mSRT, beat-by-beat values for the RR intervals (time between peak values in an electrocardiogram) will be registered using Polar RS800CX (Polar Electro) at a 1000 Hz frequency. The heart rate (HR) will serve as a control parameter for the intensity of the running test. A minimum of 48h delay will be applied between sessions 1 and 2.

At the start of session 2 ($t_1$), participants will be familiarized with the rVMA protocol (familiarization set) by performing 20 trials of non-rotated (0˚) practice trials. Afterwards, a non-rotated baseline condition set (104 trials) will be performed. After the baseline set, the EX groups (EX-DCD and EX-TD) will perform 13 minutes of intense exercise (iE) while the CON groups (i.e., CON-DCD and CON-TD) will rest during the same period of time reading or holding a conversation, with no additional exercise or musical activity permitted. Next, all participants will do an adaptation set (312 trials) in the rVMA task, with a clockwise rotation of 60˚ applied to the cardinal coordinates of the cursor movement. One hour after the end of the adaptation set, all participants will perform a 60˚ clockwise retention set (short-term retention, RT1h, 104 trials).

Sessions 3 ($t_2$) and 4 ($t_3$) will occur 24 h and 7 days from the end of the adaptation set. In each session, participants will perform a 60˚ clockwise rotated retention set (mid and long term retentions, RT24h and RT7d respectively, 104 trials each). To minimize the experimental death, researchers will collaborate with the school teachers for scheduling for appropriate recruitment and retention purposes.

## Measurements

**The rotational visuomotor adaptation task (rVMA).** During the rVMA, participants will be seated in a quiet room, in front of a 19-inch computer screen located at 1m and at eye level. With their dominant hand, participants will be asked to grasp a joystick, maintaining a claw-type grip, an elbow flexion of 90° and a comfortable shoulder position, while the forearm rests on a flat surface. The height and position of the joystick will be adjusted to meet the position criteria. The joystick movement controls a green dot on the screen (1x1cm). Individual targets will randomly appear on screen as red dots (1x1cm) in eight possible locations (45, 90, 135, 180, 225, 270, 315, and 360°) and at a radius distance of 13 cm from the center. A new target will appear every 1.5 s and will remain visible for 750 ms. Participants will be instructed to move the green dot, starting from the center of the screen, over the target (red dot) and back to the center as fast and as straight as possible in a single move. During the task a visual-motor mismatch between the joystick movement and the screen cursor movement is applied by rotating the movement of the cursor, so the greater the rotation the greater the deviation of the cursor according to the movement of the joystick (for example, see [55]). Cartesian x-y coordinates of the joystick movement and time will be registered at 120 Hz through a NI-6008 card (National Instruments Corporation).

The rVMA data fitting and reduction will be done using custom-made MATLAB R2014b programs (The MathWorks, Inc.). Cartesian x-y positions will be low-pass filtered using an eight-order dual-pass Butterworth filter with a cut-off frequency of 12 Hz. Accepted trials will have to fulfill the following conditions: startup position found within 20% of the center-to-target distance, and travelled distance equal or higher to 90% of the center-to-target distance. Movement onset of the accepted trials will be defined as the nearest point in an outward movement equal to a 10% of the center-to-target distance. Movement offset will be defined as the first point where speed decreased to a 10% of the max speed value. The adaptation set will be divided in epochs of 8 trials for analysis purposes.

**The intense exercise bout (iE).** The intense exercise (iE) bout will consist of a 13-minute 20-meter shuttle run. During this exercise bout, two speeds, based on a percentage of the estimated $VO_2$max, will be combined: a fast-paced speed (85% of $VO_2$max) and a slow-paced speed (60% $VO_2$max). A total of three series of three minutes of the fast-paced speed will be carried, interspersed with two series of two minutes of the slow-paced speed. Prior to the iE start, a warm-up protocol consisting of 2 min slow and 1 min fast will be done with the objective to familiarize the participants with the iE speeds. A 5-minute rest period will be guaranteed before starting the iE. Transition time between iE and rVMA will be 4 minutes. Participants' hearth rate will be captured following the same procedure as described for the 20mSRT.

Data of the 20mSRT will be used to characterize the participants' fitness level and to define the intensity of the exercise during iE performance in the second session. Individual maximum velocity (Vmax, km/h) will be calculated using the last completed stage (Emax) and $VO_2$max will be estimated based on the participant Vmax and age ($y$, years) [105]. Individual heart rate (HR) will be monitored by the Polar RS800CX (Polar Electro) and it will be synchronized with the time of the 20mSRT test (session 1) and with the iE intervals (session 2). We will use the theoretical maximal HR during childhood and adolescence periods (typically above 200 bpm) as a reference to assess the maximal effort level in the last part of the 20mSRT [106]. In addition, the mean HR during the iE intervals (85% and 60% of the estimated $VO_2$max) will be calculated (HR-85%$VO_2$max and HR-65%$VO_2$max). These values will be used to further characterize the intensity of the exercise during the iE.

**Optical neuroimaging: The functional Near-Infrared Spectroscopy (fNIRS).** During sessions 2, 3, and 4, a 27-channel fNIRS instrument (Brite MKII, Artinis Medical Systems,

Netherlands) will be placed on the head of the participants to record the hemodynamic changes of their right and left dorsolateral pre-frontal cortex (DLPFC) and ventrolateral pre-frontal cortex (VLPFC) to cover motor, perceptual, and attentional functionally related areas. The placement of the optodes and the cap will be anchored to the established landmarks in the 10–20 EEG electrode placement system [107] so the brain areas of interest will be targeted. In addition, measurements along one short-separation channel (SSC) on each hemisphere will be performed. The two SSCs will measure extracerebral signals (i.e., blood pressure waves, Mayer waves, respiration and cardiac cycles) which will be useful to isolate the brain hemodynamic response [108–110]. Spring-loaded grommets will be used to improve data quality in dark thick hair. Participants will be asked to rest seated during 90 seconds before the familiarization, baseline, and adaptation sets (session 2), the RT24h (session 3), and the RT7d (session 4) to record a reference resting state in the fNIRS signals before to start the rVMA task. Oxysoft software (v6.0, Artinis Medical Systems, Netherlands) will allow real-time assessment of the quality of the fNIRS signals and zero baseline will be set when the acceptable signal-to-noise ratio is obtained. Data acquisition sampling rate will be set to 50 Hz. While the fNIRS device is registering, a pulse oximeter (CMS70A Portable Pulse Oximeter, Contec Medical Systems Co., China) placed on the participant's fingertip will collect data from the general blood circulation (i.e., $SpO_2$ and heartbeat). These data will permit to characterize participants systemic responses during the rVMA task (after performing exercise or not) and, at the same time, will be another source of information to subtract the physiological noise from the fNIRS measurements [111].

Brain hemodynamic responses (oxyhemoglobin ([$0_2$Hb]), deoxyhemoglobin ([HHb]) will be measured by fNIRS during all motor task phases to assess short and long-term changes in cortical processing. A subject-specific differential path-length factor (DPF) based on the age of the participant [112, 113] will be used for this conversion. SCC data, pulse oximeter data, and low-pass filter (typically a cut-off frequency of 0.1 Hz) will be used to avoid excessive frequencies originating from the normal physiological activity, the noise from respiration, and heart beats or the movements. Then, filtered data will be normalized relative to the individual values found in the reference resting-state measures. All the processes previous to calculating the variables will be computed using custom-made programs.

## Variables

The movement during the rVMA task will be described by the calculated movement time (ms), travel distance (cm) and reaction time (ms) variables. Reaction time will be defined as the time between the target appearance and the movement onset. The movement output error will be measured through the initial directional error (IDE, deg) and root mean square error (RMSE, cm) variables. IDE will be calculated as the absolute angular difference between the ideal trajectory, a linear vector from the center to the target, and the early real trajectory, defined by the linear vector from the center to the cursor position at the time of 80 ms after the movement onset. IDE will be used as a measure of the rotation adaptation or motor consolidation avoiding the possible trajectory correction through perceptual feedback [55, 89, 92]. RMSE will be calculated to represent the straightness of the movement between the ideal trajectory and the real joystick trajectory following the procedure described in Contreras-Vidal et al. [92]. Movement straightness (i.e., RMSE) includes the initial deviation from the ideal trajectory (IDE) and the remaining movement trajectory, which could be corrected using perceptual feedback. All these variables will be calculated for each set (baseline, adaptation, RT1h, RT24h, and RT7d). Considering a large inter-subject variability usually presented by the children, the computed mean adaptation and retention variables will be normalized by subtracting the participant's mean baseline values.

During the adaptation set, the data typically present an initial rapid-error decay followed by a slower decline. As seen in other studies [55, 89, 114], these data from adaptations seem to be best fitted by a double exponential function. The initial rate of learning (RL) will be computed, as described in Coats et al. [115], as the first derivative of the first half of the function and evaluated at epoch 1 for both error variables IDE (RL-IDE) and RMSE (RL-RMSE). All individual exponential functions will be visually inspected for the plateau to assess whether the learning is achieved.

Normalized changes in concentration levels of hemodynamic variables during different sets will be analyzed. Neural activation of each cortical area (i.e., DLPFC and VLPFC) will be expressed as a relative increase of $[O_2Hb]$ and decrease in $[HHb]$ [116]. Qualitative and quantitative analysis of the changes in these variables across trials will be performed to establish differences in hemodynamic patterns.

## Data management plan

Treatment of the data will follow the Regulation (EU) 2016/679 of the European Parliament and the Council of 27 April 2016 on the protection of natural persons with regard to the processing of personal data and on the free movement of such data. The data management delegate from the Department of Education of the Generalitat de Catalunya supervised and approved the Data Protection Impact Assessment (DPIA); the data management plan was elaborated following this DPIA to ensure data protection and their proper use.

Children's parents or legal guardians will be fully informed of: (1) all the details of the study; (2) all the procedure followed to maintain confidentiality and to protect the personal data collected from them and their child; (3) the unique use of the information collected (data, pictures and videos) exclusively to scientific purpose; and (4) their rights as legal guardian of the participating child to access all his/her data, request the rectification of inaccurate data, request its deletion, limit its processing, oppose and withdraw consent for its use for certain purposes, and to withdraw at any time from part or all of the study without stating the cause and without consequences.

Participants' personal information and images will always be kept safe in order to preserve their anonymity. That is, participants' identity information will only be stored in a unique and encrypted safe file created by the researchers in charge of the project and, afterwards, participants identity will be pseudonymized though all data registration and treatment procedures. The pseudonymized data sets will be shared with the research team members to evaluate the goodness of the data with the purpose of producing the scientific publications. Only after finishing the study and with the express authorization of the Department of Education of the Generalitat de Catalunya, the data processed and anonymized will be stored in a specialized repository for research purposes. The data in this repository will be placed by default under the Creative Commons CC BY-NC-SA license and they will only be used for scientific purposes and after requesting permission from the authors (who will check the reasons for using the data) and under the same conditions that we have established.

## Data analysis

Age, sex at birth, BMI, handedness, physical activity engagement, and fitness level (estimated $VO_2max$) measures will be explored through two-way analysis of variance (ANOVA) during the allocation to evaluate if the partial-within cohort randomization created group differences. Non-parametric tests will be used when necessary; concretely, cross-tabulations will be conducted to assess group differences for sex at birth and handedness (categorical data), while group differences for physical activity engagement (ordinal data) will be analyzed using a Kruskal-Wallis test.

Descriptive statistics of all the scores/results obtained from the questionnaires answered by the parents and teachers, the anthropometric measures (height, weight, and BMI), and the tests conducted in sessions 1 (20mSRT) and 2 (TONI-4 and M-ABC2) will be used to characterize the four groups of children (O1.1). Also, 2 (Cohort) x 2 (Exercise) ANOVAs will be applied to contrast the differences between groups (CON-DCD, CON-TD, EX-DCD, and EX-TD) (O1.1), except for questionnaires, TONI-4, and M-ABC2 (ordinal data) when Kruskal-Wallis will be used.

To enable contrast and association between behavioral and cortical activity data, the adaptation set will be divided into three-subsets (AD1, AD2, and AD3) with time durations equal to the retention sets length. Variables obtained from the rVMA will be used to describe and compare the motor learning ability of children with and without DCD while performing rVMA (O1.2) and the effect of acute exercise on this motor learning (O1.3), while the data from the fNIRS will explore patterns of brain activation while learning a motor adaptation task in children with and without DCD and the effect of iE (O2). Differences between the children groups (O1.2 and O1.3) on the motor learning and on the cortical activity (O2.1 and O2.2) through all sets (AD1, AD2, AD3, RT1h, RT24h, and RT7d) will be explored using a General Lineal Mixed Model (GLMM) with repeated measures. Association of the changes in brain activation variables with the changes in the rVMA variables (O2.3) will be explored using bivariate correlations for discrete variables. In addition, group average curves for each set will be calculated for the behavioral and cortical activity data; then, cross-correlation analysis will be performed to characterize the group relationship between brain activity and behavior by set.

Assumptions for the statistical tests will be checked before their application and necessary adjustments, corrections or supplementary calculations will be applied to increase the robustness of the results. Thus, before ANOVAs and Pearson bivariate correlations, normality of the distribution will be tested for all variables via exploration of histograms, Q-Q plots and with the Shapiro-Wilk's normality test. When the normality assumption fails, variable transformation or non-parametric alternative tests will be adequately conducted. In all ANOVAs, Greenhouse-Geisser sphericity-corrected values will be used when appropriate. Variables with initial group differences and significantly correlated with the dependent variable will be considered as potential covariates in the ANOVA analyses. Bonferroni *post hoc* analyses will be performed if significant differences are found. In addition, the effect size will be calculated using $\eta^2p$ (0.01 small, 0.06 medium, 0.14 large effect). Statistical significance will be set at $p<0.05$ for all analyses. In the case of the GLMMs, Hessian matrix and G matrix will be assessed, and adequate solutions will be applied for each particular case to ensure the model suitability. In addition, Pearson residual normality analyses will be conducted to verify the suitability of the model and results obtained from GLMMs. On the other hand, auto-correlation of the behavioral and brain activity data will be evaluated before the cross-correlations analysis.

## Ethical considerations

Children's parents or legal guardians will be fully informed of all the details of the study by a trained researcher. They will sign a consent form prior to their child's participation as well as children participants will sign an assent form. In addition, as stated in the assent and consent formularies, participants will be free to quit the study in any moment if they don't want to continue participating. All the procedures of the present research project will be conducted according to the latest revision of the Helsinki Declaration. The study was approved by the Ethic Committee of Clinical Research of the Catalan Sport Administration on May 2nd, 2022 (008/CEICGC/2022).

Previous research has examined the effect of exercise on human cognition and how individual characteristics as fitness level, age, and sex could moderate the exercise effects on cognition. Labelle et al. [91] found that exercise effects on cognition were not moderated by sex. Studies where exercise has been observed to stimulate motor learning in children included male and female in the groups [55, 57]. Therefore, during the execution of this research no distinction for sex or ethnic characteristics will be made during participant recruitment. Yet, a greater number of male participants is expected because DCD has shown greater prevalence in male than female children. More male DCD probable cases are expected in the present study, compared to female DCD cases. Therefore, sex balance across groups may or may not be accomplished in this project. Despite no sex differences expected, we will introduce sex as covariate in the statistical procedures to better know if any sex effect occurred. Dissemination and transfer of the research will follow high standards in preserving the gender dimension and explaining the similarities and the differences, if any.

Lastly, a supervisor will be designated in each school in order to ensure that all study procedures are performed according to this ethics declaration. In addition, the school supervisor will be requested to provide support to the researchers in case of accident or injury of the participants during the experimental procedure, providing assistance in agreement with the emergency protocols of the school. The school supervisor will also be the mediator between researchers and participants/participants' guardians in any case of necessity.

## Discussion

According to World Health Organization [83], 5–6% of children in North America and Europe have DCD, although prevalence estimates vary between 2–19% depending on the criteria used [38, 93–95]. Children with DCD are characterized by a significant delay in the acquisition of gross and fine motor skills and impairment in the execution of coordinated motor skills [1]. These motor coordination problems have been associated to other characteristics like lower fitness level [5, 6], physical activity engagement [117, 118], cognitive and academic performance [3, 119] and poorer socio-emotional status [17]. In addition, children with DCD have also been observed to have difficulties with motor learning and not only motor control, presenting slower adaptations and worse consolidation in comparison to TD children [45, 46, 120]. However, the existing evidence is scarce and inconclusive, especially given that consolidation of motor memory has been weakly examined [35]. In summary, children with DCD present various issues that could affect their health and quality of life. In fact, without intervention, it is estimated that nearly 75% of children with DCD continue to have difficulties as adults [121]. Therefore, early identification of DCD and a timely intervention in case of this disorder seem crucial.

Acute physical exercise has been proven to improve motor learning in children [55–57]. Given the available research evidence in the literature [52–54], it seems reasonable to think that acute exercise may also have benefit on children with DCD. To our knowledge, no studies have examined this question. Knowledge about the effect of acute exercise on motor adaptation and consolidation in children with DCD may guide physical exercise interventions to improve their motor learning. Knowledge of exercise effects on learning may also guide short physical exercise interventions ("energizers") as a future technique to enhance learning and attention in the classroom. Previous researchers have successfully implemented such technique in TD children, were improvements in attention and academic performance have been observed [122–124]. Because children with DCD have been described as less attentive [125, 126], and exercise enhances attention [82, 127, 128], exercise may not only improve motor learning but also other forms of learning. In fact, school-related learning is associated with

executive functions such as attention [129, 130] but also with motor learning, such as adaptation and consolidation of visuo-motor integration tasks. Therefore, there is a close relationship between the development of the brain substrates responsible for motor learning and those of executive functions [131]. Because the difficulties in motor control and learning in DCD children have been related to various health and quality of life issues, improving their learning and attention through short exercise bouts could carry a much larger impact in the lives of children with DCD.

Moreover, although that exercise effects have been related to an increases in some neuro-chemicals (catecholamines and BDNF) [132–134] little is known about the possible mechanisms underlying the exercise benefits on motor learning. Therefore, studies where concurrent brain imaging and behavioral assessments are used would be the next step to explore the exercise benefits on learning in children. To our knowledge, there are no such studies, and certainly not in children with DCD.

In summary, the present research project will add new evidence regarding the characteristics of DCD. Because both processes during learning of a new motor skill (i.e., adaptation and consolidation) will be analyzed, this project will also add new evidence on how children with DCD adapt to the visuo-motor integration tasks, how they consolidate their motor memory and how they retrieve the learned skill in comparison to TD children. In addition, this project will present prime evidence regarding the effect of acute exercise on the learning ability of children with DCD. Lastly, this project will utilize fNIRS as the device to assess learning in children with and without DCD. Providing such knowledge will constitute an important advancement not only in better characterizing children with DCD and establishing acute exercise as a potential intervention to improve motor learning, but also in designing educational materials for educators and families related to children with DCD. One way that we may help teachers and parents in identifying those children at risk or high probability of presenting DCD by generating and distributing (via workshops) guidelines for the diagnosis of this disorder and proper interventions for these children, which will have to include robust conclusions from our project but also its inherent limitations to prevent overreaching the conclusions summarized in the guidelines. At the end of the project, we will share the results emphasizing the factors that could predict DCD and the effects of acute exercise on learning in children with and without DCD including a plausible neurological mechanism for the differences between children with DCD and those with TD.

## Supporting information

**S1 Checklist. SPIRIT 2013 checklist recommended items to include in a clinical trial protocol.**
(DOC)

**S1 File.**
(PDF)

## Acknowledgments

We thank Dr. Priscila Tamplain (Developmental Motor Cognition Lab, Department of Kinesiology, University of Texas at Arlington, USA) and Dr. Nadja Schott (Institute of Sport and Exercise Science, University of Stuttgart, Germany) for their collaboration in the project. We also thank the Grup de Recerca en Activitat Física, Alimentació i Salut (GRAFAiS, Generalitat de Catalunya 2021SGR/01190) for their administrative support.

## Author Contributions

**Conceptualization:** Albert Busquets, Blai Ferrer-Uris, Rosa Angulo-Barroso.

**Formal analysis:** Albert Busquets, Blai Ferrer-Uris, Faruk Bešlija, Rosa Angulo-Barroso.

**Funding acquisition:** Albert Busquets, Blai Ferrer-Uris, Rosa Angulo-Barroso.

**Investigation:** Albert Busquets, Blai Ferrer-Uris, Manuel Añón-Hidalgo, Rosa Angulo-Barroso.

**Methodology:** Albert Busquets, Blai Ferrer-Uris, Turgut Durduran, Faruk Bešlija, Rosa Angulo-Barroso.

**Project administration:** Albert Busquets, Rosa Angulo-Barroso.

**Software:** Blai Ferrer-Uris, Turgut Durduran, Faruk Bešlija.

**Supervision:** Albert Busquets, Rosa Angulo-Barroso.

**Writing – original draft:** Albert Busquets, Blai Ferrer-Uris, Rosa Angulo-Barroso.

**Writing – review & editing:** Albert Busquets, Blai Ferrer-Uris, Turgut Durduran, Faruk Bešlija, Manuel Añón-Hidalgo, Rosa Angulo-Barroso.

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
