## [Decision Letter · Decision Letter 0]

23 Oct 2023

PONE-D-23-24057Study protocol to examine the effects of acute exercise on motor learning and brain activity in children with developmental coordination disorder (ExLe-Brain-DCD)PLOS ONE

Dear Dr. Busquets,

Thank you for submitting your manuscript to PLOS ONE. After careful consideration, we feel that it has merit but does not fully meet PLOS ONE’s publication criteria as it currently stands. Therefore, we invite you to submit a revised version of the manuscript that addresses the points raised during the review process.

You will find comments from a reviewer below. I included a few additional comments below. In the resubmission, please reply to all comments made by me and the reviewer. Objective 1.2 and 1.3 (line 351 and onward) – the wording here is difficult to follow. Specifically the phrase “while performing…”. In objective 1.2, the assessment of motor learning is done with this task so perhaps rephrase to “Compare motor learning (adaptation and consolidation) ability in children with and without DCD in a rotation visuo-motor task”. Similar changes can be made for Objective 1.3. I strongly encourage the authors to refrain from stating global descriptors such as “visuomotor integration and attentional network areas” as part of the objectives (Objective 2). There are many areas that would fall under these labels (e.g., parietal cortex is known for sensorimotor integration and fronto-parietal networks are critically involved in attentional processes). Note also prefrontal cortices are involved in more than VM integration and attention (e.g., executive functioning). For clarity, it is advised to simply state the regions of interest that will be examined as part of the research. That is, readers know the specific regions examined. The portions of the manuscript that justify these regions can then include proposed functional roles. I would like to echo Reviewer 1 concerns with respect to “matching” across TD and DCD on this long list of factors. Children with DCD, on average, tend to be less active. So finding matched TD participants could add an extra burden that may compromise the completion of the research. And, if they are matched, I would imagine that this would mean the sampling of TD children is even less representative of the general population. Accordingly, it raises questions as to how generalizable the sample of TD children is. Although I recognize why this matching has been done, I simply wanted to extend Reviewer 1’s comments to highlight this issue further. Ultimately, I think a multitude of approaches can be justified here. So something for the research team to consider further before starting the experiment. Please submit your revised manuscript by Dec 07 2023 11:59PM. If you will need more time than this to complete your revisions, please reply to this message or contact the journal office at plosone@plos.org. Please include the following items when submitting your revised manuscript:A rebuttal letter that responds to each point raised by the academic editor and reviewer(s). You should upload this letter as a separate file labeled 'Response to Reviewers'.A marked-up copy of your manuscript that highlights changes made to the original version. You should upload this as a separate file labeled 'Revised Manuscript with Track Changes'.An unmarked version of your revised paper without tracked changes. You should upload this as a separate file labeled 'Manuscript'.If applicable, we recommend that you deposit your laboratory protocols in protocols.io to enhance the reproducibility of your results. Protocols.io assigns your protocol its own identifier (DOI) so that it can be cited independently in the future. For instructions see: https://journals.plos.org/plosone/s/submission-guidelines#loc-laboratory-protocols. Additionally, PLOS ONE offers an option for publishing peer-reviewed Lab Protocol articles, which describe protocols hosted on protocols.io. Read more information on sharing protocols at https://plos.org/protocols?utm_medium=editorial-email&utm_source=authorletters&utm_campaign=protocols.

We look forward to receiving your revised manuscript.

Kind regards,

Bradley R. King

Academic Editor

PLOS ONE

[We thank Dr. Priscila Tamplain (Developmental Motor Cognition Lab, Department of Kinesiology, University of Texas at Arlington, USA) and Dr. Nadja Schott (Institute of Sport and Exercise Science, University of Stuttgart, Germany) for their collaboration in the project. We also thank the Grup de Recerca en Activitat Física, Alimentació i Salut (GRAFAiS, Generalitat de Catalunya 2021SGR/01190) for their support.]

 [AB, BF, FB, and RA as authors of this study that is part of the R+D+i project PID2020-120453RB-I00 received funding from the Ministerio de Ciencia e Innovación – Agencia Estatal de Invenstigación (https://www.aei.gob.es/; MCIN/AEI/10.13039/501100011033/). In addition, MA earned a PhD fellowship funded by the Institut Nacional d'Educació Física de Catalunya (INEFC) of the Generalitat de Catalunya (https://inefc.gencat.cat/es/inefc_barcelona/). The funders had and will not have a role in the study design, data collection and analyses, decision to publish, or preparation of the manuscript.]

Reviewers' comments:

Reviewer's Responses to Questions

**Comments to the Author**

1. Does the manuscript provide a valid rationale for the proposed study, with clearly identified and justified research questions?

Reviewer #1: Yes

2. Is the protocol technically sound and planned in a manner that will lead to a meaningful outcome and allow testing the stated hypotheses?

Reviewer #1: Yes

3. Is the methodology feasible and described in sufficient detail to allow the work to be replicable?

Reviewer #1: Yes

4. Have the authors described where all data underlying the findings will be made available when the study is complete?

Reviewer #1: Yes

5. Is the manuscript presented in an intelligible fashion and written in standard English?

Reviewer #1: Yes

6. Review Comments to the Author

You may also provide optional suggestions and comments to authors that they might find helpful in planning their study.

Reviewer #1: The present manuscripts presents a Study Protocol for a project that aims to unravel the underlying mechanism of the (motor) learning difficulties in children with DCD. This is a very interesting, important and ambitious topic. The methodology of the project seems sound and I am already looking forward to the results. I only have a few comments/suggestions with respect to the paper and project.

1. The introduction is rather lengthy and somewhat repetitive. The text can be condensed. E.g. I do appreciate that the authors provide a comprehensive overview of DCD and its consequences, but this information can be summarized and the reader can be referred to some excellent recent review papers. Given that the many authors refer to DSM V (APA, 2013) it seems to make sense to refer to this manuscript too.

Specific minor comments

Abstract

L39: Given that these effects are generally limited to (moderate to) vigorous physical activity I would suggest being specific here. Or to use physical exercise.

L168: typo: prefrontal cortex' role

L363 and 367: typically developing children is the preferred term (instead of developed)

L376: the groups will be matched on age, gender, weight, handedness, PA engagement and fitness level. I understand why, but it seems to be very complex to match each and every child on 6 factors. Can the authors please explain how this will be done.

L421: Given that the study will use matched control groups, allocation cannot be fully randomized, I think.

L602: What are the regions of interest and how will parcellation of the regions of interest be done?

L636-638: Is this analysis necessary, given that the groups will be matched?

L699: Is it necessary to use sex as a covariate, given that the groups will be matched? Also, please note that the construct "gender" was used before.

7. PLOS authors have the option to publish the peer review history of their article (what does this mean?). If published, this will include your full peer review and any attached files.

Reviewer #1: No

---

## [Author Response · Author response to Decision Letter 0]

14 Nov 2023

Responses to Editor:

1. Objective 1.2 and 1.3 (line 351 and onward) – the wording here is difficult to follow. Specifically the phrase “while performing…”. In objective1.2, the assessment of motor learning is done with this task so perhaps rephrase to “Compare motor learning (adaptation and consolidation) ability in children with and without DCD in a rotation visuo-motor task”. Similar changes can be made for Objective 1.3. 

I strongly encourage the authors to refrain from stating global descriptors such as “visuomotor integration and attentional network areas” as part of the objectives (Objective 2). There are many areas that would fall under these labels (e.g., parietal cortex is known for sensorimotor integration and fronto-parietal networks are critically involved in attentional processes). Note also prefrontal cortices are involved in more than VM integration and attention (e.g., executive functioning). For clarity, it is advised to simply state the regions of interest that will be examined as part of the research. That is, readers know the specific regions examined. The portions of the manuscript that justify these regions can then include proposed functional roles. 

Thank you for pointing out this issues and making some rephrasing proposals. Indeed, we agree with the editor that the objectives were not easy to read. We also understand and agree to your point about avoiding global descriptors and use instead specific cerebral areas. We have addressed all these issues in the revised version. We hope the introduced changes provide a clearer and easier understanding of our aims. 

Please, see changes in lines 369 to 381 in the Revised Manuscript with Track Changes.

2. I would like to echo Reviewer 1 concerns with respect to “matching” across TD and DCD on this long list of factors. Children with DCD, on average, tend to be less active. So finding matched TD participants could add an extra burden that may compromise the completion of the research. And, if they are matched, I would imagine that this would mean the sampling of TD children is even less representative of the general population. Accordingly, it raises questions as to how generalizable the sample of TD children is. Although I recognize why this matching has been done, I simply wanted to extend Reviewer 1’s comments to highlight this issue further. Ultimately, I think a multitude of approaches can be justified here. So something for the research team to consider further before starting the experiment. 

Since both, the editor and the reviewer, have raised their concerns regarding the proposed criteria for group matching in our study, we have reflected on its viability. After careful reconsideration, we agree that having so many matching criteria would be very difficult to handle. Therefore, we have modified our proposal so only three criteria will be used to match participant distribution among groups: age, sex, and handedness. Therefore, BMI (not weight), physical activity engagement, and physical fitness will be used as covariates if needed. That is, these three variables will be compared among groups (as well as age, sex, and handedness) to explore any group distribution difference and used as covariates in case group differences are found.

We want to thank the editor and the reviewer for this comment, as we think this is a very appropriate change to our study protocol. The initially proposed group matching could have possibly resulted in an unpractical and very difficult (or impossible) approach. 

Please, see the modified sentence as lines 390-391 in the Revised Manuscript with Track Changes.

1. Please ensure that your manuscript meets PLOS ONE's style requirements, including those for file naming. The PLOS ONE style templates can befound at

Done.

When you resubmit, please ensure that you provide the correct grant numbers for the awards you received for your study in the ‘FundingInformation’ section.

Thank you to noticing this issue, we changed the text of the ‘Funding Information’ to match the text included in the ‘Financial Disclosure’ section (please, see the Cover letter). 

The ’Ethics Statement’ was already at the end of the ‘Materials and methods’ section (lines 697-727 in the Revised Manuscript with Track Changes). Please, let us know if it has to be moved anywhere else in the manuscript.

[We thank Dr. Priscila Tamplain (Developmental Motor Cognition Lab, Department of Kinesiology, University of Texas at Arlington, USA) and Dr. NadjaSchott (Institute of Sport and Exercise Science, University of Stuttgart, Germany) for their collaboration in the project. We also thank the Grup deRecerca en Activitat Física, Alimentació i Salut (GRAFAiS, Generalitat de Catalunya 2021SGR/01190) for their support.]

We note that you have provided funding information that is not currently declared in your Funding Statement. However, funding information shouldnot appear in the Acknowledgments section or other areas of your manuscript. We will only publish funding information present in the FundingStatement section of the online submission form.

The support of the Grup de Recerca en Activitat Física, Alimentació i Salut (GRAFAiS, Generalitat de Catalunya 2021SGR/01190) was only administrative. We clarified that in the ‘Acknowledgments’ sections (line 808 in the Revised Manuscript with Track Changes).

Please remove any funding-related text from the manuscript and let us know how you would like to update your Funding Statement. Currently, yourFunding Statement reads as follows:

[AB, BF, FB, and RA as authors of this study that is part of the R+D+i project PID2020-120453RB-I00 received funding from the Ministerio de Cienciae Innovación – Agencia Estatal de Invenstigación (https://www.aei.gob.es/; MCIN/AEI/10.13039/501100011033/). In addition, MA earned a PhDfellowship funded by the Institut Nacional d'Educació Física de Catalunya (INEFC) of the Generalitat de Catalunya(https://inefc.gencat.cat/es/inefc_barcelona/). The funders had and will not have a role in the study design, data collection and analyses, decision to publish, or preparation of the manuscript.]

We revised the manuscript looking for any funding-related text. No section contains funding-related texts in the new version of the manuscript. We removed the funding information from the Title page of the manuscript and now it is available in the cover letter. 

5. Please review your reference list to ensure that it is complete and correct. If you have cited papers that have been retracted, please include therationale for doing so in the manuscript text, or remove these references and replace them with relevant current references. Any changes to thereference list should be mentioned in the rebuttal letter that accompanies your revised manuscript. If you need to cite a retracted article, indicate thearticle’s retracted status in the References list and also include a citation and full reference for the retraction notice.

Done.

Responses to reviewers:

1. The introduction is rather lengthy and somewhat repetitive. The text can be condensed. E.g. I do appreciate that the authors provide a comprehensive overview of DCD and its consequences, but this information can be summarized and the reader can be referred to some excellent recent review papers. Given that the many authors refer to DSM V (APA, 2013) it seems to make sense to refer to this manuscript too.

We agree with the reviewer that the introduction is long and some parts provide too much, and somewhat redundant, detail. In agreement with this comment, we have made an effort to shorten the introduction section, summarizing those parts where too much detailed information was provided, and deleting some sentences that were not critical for the justification of our study protocol.

We hope the implemented changes improve the focus of the introduction and provide a sufficiently comprehensive review of the literature background of our research protocol.

Please, see the introduction section of the revised manuscript were many changes have been performed in each of the pre-existing paragraphs.

Specific minor comments

AbstractL39: Given that these effects are generally limited to (moderate to) vigorous physical activity I would suggest being specific here. Or to use physical exercise.

We agree with the reviewer. We have detailed that the effects are specific to moderate-to-vigorous physical exercise.

The new sentence is (lines 40-41 in the Revised Manuscript with Track Changes): “Moderate-to-vigorous physical exercise has been proven to improve motor learning (adaptation and consolidation) in children with or without disorders.”

L168: typo: prefrontal cortex' role

Thank you for noticing this typo. We have corrected it in the revised version of the manuscript. See line 175 in the Revised Manuscript with Track Changes.

L363 and 367: typically developing children is the preferred term (instead of developed)

Thank you for noticing this inconsistency. We have changed these two lines to be consistent with the most appropriate term (typically developing). 

L376: the groups will be matched on age, gender, weight, handedness, PA engagement and fitness level. I understand why, but it seems to be very complex to match each and every child on 6 factors. Can the authors please explain how this will be done.

Please, see our comment to the editor in point 2 where we address this concern. Thank you for this important contribution to our protocol. Please, see the modified sentence as lines 390-391 in the Revised Manuscript with Track Changes.

L421: Given that the study will use matched control groups, allocation cannot be fully randomized, I think.

We agree with the reviewer. In our case, participant allocation will be partially randomized, not fully randomized. We have clarified this issue in the revised manuscript. Please, see the changes performed in lines 52 and 465-466 in the Revised Manuscript with Track Changes.

L602: What are the regions of interest and how will parcellation of the regions of interest be done?

Thank you for this comment, which has also been raised by the editor. We have modified the manuscript to specify that the two brain areas of interest are the dorsolateral pre-frontal cortex (DLPFC) and the ventrolateral pre-frontal cortex (VLPFC). Since we have a limitation of 27 channels in our current fNIRS system, we need to focus on the most relevant brain areas for our aims, population and experimental protocol, that is, the pre-frontal cortex. We also specifically indicate that the established 10-20 EEG protocol will be followed for the optodes placements so the targeted brain areas can be evaluated.

See lines 556-562 in the Revised Manuscript with Track Changes.

L636-638: Is this analysis necessary, given that the groups will be matched?

We understand the reviewer’s doubts on this statistical procedure, but we still think that an analysis is necessary for the following reason. Comparison of the group matching criteria (age, sex, and handedness) will be performed to corroborate that the participant allocation was performed correctly. Furthermore, since now we will not use BMI, physical activity engagement, and physical fitness as group distribution criteria, we need to check if there exist any group effect on these criteria and use them as covariates whenever necessary.

L699: Is it necessary to use sex as a covariate, given that the groups will be matched? Also, please note that the construct "gender" was used before.

Thank again for this comment. In this case the proposed analysis by sex reflects our interest to address an ethical consideration that although our efforts to balance groups by sex, there might still be a sex difference in some of our measures and we would like to know the answer to this question.

We have revised the manuscript to be consistent with the terminology “sex” to refer to the sex designation at birth, and the use of gender to address a more general term that could include sex at birth of other forms of defining “sex” designation.

---

## [Decision Letter · Decision Letter 1]

30 Jan 2024

PONE-D-23-24057R1Study protocol to examine the effects of acute exercise on motor learning and brain activity in children with developmental coordination disorder (ExLe-Brain-DCD)PLOS ONE

Dear Dr. Busquets,

Thank you for submitting your manuscript to PLOS ONE. I want to apologize for the delay in getting back to you after the latest manuscript submission. As stated in our earlier correspondence, the policy of PLOS One for study protocols is to have a statistical reviewer. We had difficulty getting a qualified individual to take on this review but we just received comments from such an individual. You will see their comments and suggestions below. Accordingly, we invite you to submit a revised version of the manuscript that addresses these concerns and comments. I have included the full review below, but I wish to highlight here what I consider the most important comments that warrant your attention.  - The comment on the choice of the control group as it pertains to the design, effect size and size of the proposed sample. - The comment on the type of data (potentially ordinal) and the choice of statistical test (parametric vs. non parametric). - The comment on the data analysis with respect to the number of groups and the type of test used (i.e., ANOVA vs. t-tests). And the related comments with respect to clarity of the design (e.g., clarifying number of observations)

We look forward to receiving your revised manuscript.

Kind regards,

Bradley R. King

Academic Editor

PLOS ONE

Journal Requirements:

Reviewers' comments:

Reviewer's Responses to Questions

**Comments to the Author**

1. Does the manuscript provide a valid rationale for the proposed study, with clearly identified and justified research questions?

Reviewer #2: Partly

2. Is the protocol technically sound and planned in a manner that will lead to a meaningful outcome and allow testing the stated hypotheses?

Reviewer #2: No

3. Is the methodology feasible and described in sufficient detail to allow the work to be replicable?

Reviewer #2: No

4. Have the authors described where all data underlying the findings will be made available when the study is complete?

Reviewer #2: Yes

5. Is the manuscript presented in an intelligible fashion and written in standard English?

Reviewer #2: No

6. Review Comments to the Author

You may also provide optional suggestions and comments to authors that they might find helpful in planning their study.

Reviewer #2: Important note: This review pertains only to ‘statistical aspects’ of the study and so ‘clinical aspects’ [like medical importance, relevance of the study, ‘clinical significance and implication(s)’ of the whole study, etc.] are to be evaluated [should be assessed] separately/independently. Further please note that any ‘statistical review’ is generally done under the assumption that (such) study specific methodological [as well as execution] issues are perfectly taken care of by the investigator(s). This review is not an exception to that and so does not cover clinical aspects {however, seldom comments are made only if those issues are intimately / scientifically related & intermingle with ‘statistical aspects’ of the study}. Agreed that ‘statistical methods’ are used as just tools here, however, they are vital part of methodology [and so should be given due importance]. I look at the manuscript in/with statistical view point, other reviewer(s) look(s) at it with different angle so that in totality the review is very comprehensive. However, there should be efforts from authors side to improve (may be by taking clues from reviewer’s comments). Therefore, please do not limit the revision only (with respect) to comments made here.

COMMENTS: There are quite a few issues (including few serious objections/observations) about which I have different opinion. Such observations/concerns are given below:

Firstly, I observed that your ABSTRACT [though drafted alright in my opinion], is ‘assay type’. It is preferable to divide the ABSTRACT with small sections like ‘Objective(s)’, ‘Methods’, ‘Results’, ‘Conclusions’, etc. which is an accepted practice of most of the good/standard journals [including this one, though ‘The PLoS One Guidelines to Authors’ did not specify an Abstract format, it is desirable]. It will definitely be more informative then, I guess, whatever the article type may be (including ‘protocol’).

As noted by some reviewer earlier, the introduction is lengthy [nearly occupies 38% of total length though I appreciate providing a comprehensive overview of DCD and its consequences]. Nearly 65% references (out of total 140) are related to the introduction. Can this be reduced further? (Assumed that you might have done it earlier). Further, I request these learned authors to kindly recheck the ‘Sample size calculation’ [lines 357-368] because {though ‘GPower’ is an excellent package} the results are different than well-known standard [table-2 on page 158 of Jacob Cohen’s paper “A power primer” in Psychological Bulletin, 1992, vol.:112, pp 155-159 (which is a sort of summary of the excellent book by Cohen himself titled ‘Statistical power analysis for the behavioral sciences’, Academic Press, 1977, New York)]. This is probably due to large ‘effect size’ assumed {The effect sizes (ES) in these studies were 0.78 and 0.55} whereas in fact for such intervention(s) the effect size found/noted is generally at-the-most be ‘medium’.

I have one basic doubt regarding the correctness of choice of a control group [for example, look at the statement made in lines 246-7: whether the children with DCD could benefit from this strategy to improve their motor learning capacity is not known]. From the account given in lines 44 to 46 {One hundred twenty children will be recruited (60 DCD, 60 controls) and within-cohort randomly assigned to either exercise (13-minute shuttle run task) or rest prior to performing the rVMA task} it is clear that cohort of 60 children with DCD will further be divided in two groups which is alright. But then I doubt about the ‘effect size’ assumed (very large & so the required n=120). Therefore, I again request authors to recheck the control group of those studies {line 357-359: Statistical power analyses were performed for the sample size estimation, based on initial directional error (IDE) data from Kagerer et al. [47] study (N=20) comparing TD to DCD and from Ferrer-Uris et al. [57] study (N=21) comparing EX and CON groups}. ‘What exactly you mean by “Statistical power analyses were performed for the sample size estimation”. Do we usually perform ‘Statistical power analyses for the sample size estimation’?

What is in lines 584-6 [Assumptions for the statistical tests will be checked before their application and necessary adjustments, corrections or supplementary calculations will be applied to increase the robustness of the results] is highly appreciated, however, I request authors to carefully read the following note which is pasted from one famous standard textbook on ‘Medical Research Methodology’ [though I am sure that the authors already know these things].

Please use suitable non-parametric test(s)/technique(s) while dealing with data that are in ‘ordinal’ level of measurement even if [despite that] the distribution may be ‘Gaussian’. Testing ‘normality’ in sample [by using any normality test(s)} is not required/desired while dealing with data that are in ‘ordinal’ level of measurement [as most of the normality tests are not valid for ‘ordinal’ data]. Agreed that there is/are no non-parametric test(s)/technique(s) available to be used as alternative in all situation(s), but should be used whenever/wherever they are available.

Note that though the measures/tools used are appropriate, most of them are likely to yield data that are in ‘ordinal’ level of measurement [and not in ratio level of measurement for sure {as the score two times higher does not indicate presence of that parameter/phenomenon as double (for example, a Visual Analogue Scales VAS score or say ‘depression’ score)}]. Then application of suitable non-parametric (or distribution free) test(s) is/are indicated/advisable.

Description in ‘Data analysis’ section is confusing in the sense that ‘how many groups data are dealt here’? If two groups, then why do you need ANOVA? How the measures are to be explored through one-way analysis of variance (ANOVA)? [refer to lines 559-561: Age, sex at birth, BMI, handedness, physical activity engagement, and fitness level (estimated VO2max) measures will be explored through one-way analysis of variance (ANOVA)]. What do you mean by exploring measures through ANOVA? Refer to lines 576-78 where stated that “Differences between the children groups (O1.2 and O1.3) on the motor learning and on the cortical activity (O2.1 and O2.2) through all sets will be explored using a General Lineal Mixed Model (GLMM) with repeated measures”. Are differences between the children groups explored using a General Lineal Mixed Model (GLMM) with repeated measures? How many times repeated observations you have? Please clarify all the aspects of ‘DESIGN’ of the study, first [very clearly]. Remember/mind you that this is a scientific/academic document and so all details should be clearly/correctly communicated (do not take readers’ for granted). Kindly check for the ‘English’ language. Agreed that English is not our mother tongue (definitely not mine, may or may not be yours but certainly not of many readers).

In lines 685-88, it is stated that “Recognizing the inherent limitations of this project, we may help teachers and parents in identifying those children at risk or high probability of presenting DCD by generating and distributing (via workshops) guidelines for the diagnosis of this disorder and proper interventions for these children”. Is that a part of this study/project? Does that mean {according to authors} there are no other {this is not a limitation of this study anyway}, limitation(s) of this study? As pointed out in ‘important note’ above “This review pertains only to ‘statistical aspects’ of the study and so ‘clinical aspects’ should be assessed separately/independently [one should carefully consider/look at the clinical implications of the study]. In my opinion, to make this article acceptable (which is possible though not easy), some amount of re-vision (re-drafting) may be needed. However, please do not limit the revision only (with respect) to comments made here.

The respected ‘Editor’ may consider accepting/further processing only if found ‘clinical implications’ valuable [i.e., add(s) to clinical knowledge or positively influence clinical practice]. ‘Major revision’ is recommended.

7. PLOS authors have the option to publish the peer review history of their article (what does this mean?). If published, this will include your full peer review and any attached files.

Reviewer #2: No

---

## [Author Response · Author response to Decision Letter 1]

11 Mar 2024

Responses to Editor:

We really appreciate your efforts to find a statistical reviewer and highlight the most important issues to consider for this review. We did our best to answer the reviewer’s questions and to include his/her suggestions in the new manuscript version. 

To ensure we covered all the reviewer’s comments that you highlighted, we organized our answers following the topics described in the four bullets you presented to us. 

Responses to reviewers:

Firstly, I observed that your ABSTRACT [though drafted alright in my opinion], is ‘assay type’. It is preferable to divide the ABSTRACT with small sections like ‘Objective(s)’, ‘Methods’, ‘Results’, ‘Conclusions’, etc. which is an accepted practice of most of the good/standard journals [including this one, though ‘The PLoS One Guidelines to Authors’ did not specify an Abstract format, it is desirable]. It will definitely be more informative then, I guess, whatever the article type may be (including ‘protocol’).

Thank you for considering the abstract of the manuscript well drafted. Following your advice, we divided the abstract into sections but adapting the sections to the type of article (i.e., Study Protocol article) and looking at how other articles in PLoS One have done it (for example: Sun, F. et al., 2022, and Tse, A.C.Y, et al, 2022). Therefore, the abstract is divided into “Introduction”, “Objectives”, “Methods”, and “Discussion” (pages 2-3, lines 31-58 of the new manuscript version).

As noted by some reviewer earlier, the introduction is lengthy [nearly occupies 38% of total length though I appreciate providing a comprehensive overview of DCD and its consequences]. Nearly 65% references (out of total 140) are related to the introduction. Can this be reduced further? (Assumed that you might have done it earlier). 

We revised again the ‘Introduction’ section to shorten its length (see page 6 and 8 of the new manuscript version). Although the ‘Introduction’ may still appear lengthy and with many references, we think that the synthesized information we present is essential to provide the reader with the necessary reasons that justify the study and its design. 

Further, I request these learned authors to kindly recheck the ‘Sample size calculation’ [lines 357-368] because {though ‘GPower’ is an excellent package} the results are different than well-known standard [table-2 on page 158 of Jacob Cohen’s paper “A power primer” in Psychological Bulletin, 1992, vol.:112, pp 155-159 (which is a sort of summary of the excellent book by Cohen himself titled ‘Statistical power analysis for the behavioral sciences’, Academic Press, 1977, New York)]. This is probably due to large ‘effect size’ assumed {The effect sizes (ES) in these studies were 0.78 and 0.55} whereas in fact for such intervention(s) the effect size found/noted is generally at-the-most be ‘medium’. 

I have one basic doubt regarding the correctness of choice of a control group [for example, look at the statement made in lines 246-7: whether the children with DCD could benefit from this strategy to improve their motor learning capacity is not known]. From the account given in lines 44 to 46 {One hundred twenty children will be recruited (60 DCD, 60 controls) and within-cohort randomly assigned to either exercise (13-minute shuttle run task) or rest prior to performing the rVMA task} it is clear that cohort of 60 children with DCD will further be divided in two groups which is alright. But then I doubt about the ‘effect size’ assumed (very large & so the required n=120). Therefore, I again request authors to recheck the control group of those studies {line 357-359: Statistical power analyses were performed for the sample size estimation, based on initial directional error (IDE) data from Kagerer et al. [47] study (N=20) comparing TD to DCD and from Ferrer-Uris et al. [57] study (N=21) comparing EX and CON groups}. ‘What exactly you mean by “Statistical power analyses were performed for the sample size estimation”. Do we usually perform ‘Statistical power analyses for the sample size estimation’?

Answer to the comment on the choice of the control group as it pertains to the design

The study protocol presented in our manuscript has two main objectives: (1) To study acute intense exercise effects on motor learning (adaptation and consolidation) in children with and without DCD from a behavioral perspective (performance and execution), and (2) To explore brain activity in the dorsolateral pre-frontal cortex (DLPFC) and ventrolateral pre-frontal cortex (VLPFC) while learning a motor adaptation task in children with and without DCD, under rest or post-exercise conditions. (Please, see the specific objectives detailed in pages 12-13 lines 276-303 of the new manuscript version). On that account, it is important to point that there are two main factors that impact the study design to fulfill these objectives: the disorder condition (if children have or not DCD), and the performance of the acute exercise intervention (if children perform or not acute intense exercise before the rotational visuomotor task).

Regarding the disorder condition, some previous articles studied the motor learning capacity in children with DCD contrasting them to typically develop children from a behavioral perspective (for example, Kagerer et al. 2004; Kagerer et al., 2006; King et al., 2011) but there are no studies to our knowledge that provided evidence of the prefrontal cortex (PFC) activity to explore possible differences in the brain responses of the DCD and TD children during a motor learning task. Furthermore, the existing research literature has not examined the PFC activity and its contribution on rotational visuomotor adaptation tasks in children.

Regarding the acute exercise effect on motor learning in children, very few articles examined the behavioral changes during adaptation and consolidation phases in children with TD (Lundbye-Jensen et al., 2017; Ferrer-Uris et al., 2018; Angulo-Barroso et al., 2019). No studies focusing on children including concurrent PFC activity and behavioral assessments have been published yet, and much less in children with DCD. 

Therefore, the inclusion of the two groups of TD children (CON-TD and EX-TD) in our study design is important to properly draw robust results not only about the impact of the acute exercise on motor learning in children with and without DCD, but also regarding the possible underlying mechanisms related to the PFC that are implicated on a visuomotor adaptation task in children. That way, and by including the four groups in our experimental design (CON-TD, CON-DCD, EX-TD, and EX-DCD), we will be able to answer the questions explicitly exposed in the specific objectives:

• CON-TD vs CON-DCD: (O1.2.) Compare motor learning (adaptation and consolidation) ability in children with and without DCD while performing a rotational visuo-motor task (rVMA); and (O2.1) assess changes in DLPFC and VLPFC activity ([02Hb]) during rVMA learning (adaptation and consolidation) in typically developing children and in children with DCD.

• CON-TD vs EX-TD, CON-DCD vs EX-DCD and EX-TD vs EX-DCD: (O1.3) Examine the effect of acute intense exercise on motor learning (adaptation and consolidation) ability in children with and without DCD while performing a rotational visuo-motor task; and (O2.2) Evaluate the impact of acute intense exercise on changes in DLPFC and VLPFC activity ([02Hb]) during rVMA learning (adaptation and consolidation) in typically developing children and in children with DCD.

Answer to the comment on the effect size and size of the proposed sample

Power analysis is normally conducted before the data collection. The main purpose underlying power analysis is to help the researcher to determine the smallest sample size that is suitable to detect the effect of a given test at the desired level of significance. Therefore, power analysis is a critical step for study design to determine the appropriate sample size. This process requires the determination of effect sizes from relevant publications. 

To conduct the power analysis and obtain a recommended sample size regarding comparisons between TD and DCD or between EX and CON, we used initial directional error (IDE) data from Kagerer et al. (2006) and Ferrer-Uris et al. (2018) (both studies mentioned in the previous point since the former utilized children with DCD and the later included acute exercise impact on motor learning in TD children). 

As the reviewer suggested, we recalculated the effect sizes and estimated the sample size following Cohen’s paper (Cohen, 1992). In the next table, the reviewer can see all the data used to calculate the effect size values (now expressed as Cohen’s d) and the sample size estimated using both methods GPower and Cohen’s tables. The results of the sample size estimation are similar despite the method used. (Please see also table included in the document 'Response to reviewers').

To clarify the calculations made about the sample size and ensure their reproducibility using GPower or the Cohen’s publications, we included the data showed in the previous table in the ‘Sample size calculation’ sub-section of the new version of the manuscript (page 15-16, lines 366-375).

What is in lines 584-6 [Assumptions for the statistical tests will be checked before their application and necessary adjustments, corrections or supplementary calculations will be applied to increase the robustness of the results] is highly appreciated, however, I request authors to carefully read the following note which is pasted from one famous standard textbook on ‘Medical Research Methodology’ [though I am sure that the authors already know these things].

Please use suitable non-parametric test(s)/technique(s) while dealing with data that are in ‘ordinal’ level of measurement even if [despite that] the distribution may be ‘Gaussian’. Testing ‘normality’ in sample [by using any normality test(s)} is not required/desired while dealing with data that are in ‘ordinal’ level of measurement [as most of the normality tests are not valid for ‘ordinal’ data]. Agreed that there is/are no non-parametric test(s)/technique(s) available to be used as alternative in all situation(s), but should be used whenever/wherever they are available.

Note that though the measures/tools used are appropriate, most of them are likely to yield data that are in ‘ordinal’ level of measurement [and not in ratio level of measurement for sure {as the score two times higher does not indicate presence of that parameter/phenomenon as double (for example, a Visual Analogue Scales VAS score or say ‘depression’ score)}]. Then application of suitable non-parametric (or distribution free) test(s) is/are indicated/advisable.

Answer to the comment on the type of data (potentially ordinal) 

Thank you for pointing out the need to indicate the non-parametric test we will use in case of ordinal data. It is important to note that all data obtained to contrast the motor learning capacity of the DCD and TD children and the effect of performing exercise (specific objectives: O1.2, O1.3, O2.1, O2.2., and O2.3) are in ratio level of measurements (concretely they are continuous data). The rotational visuomotor adaptation task (rVMA) in our protocol will be conducted in a computer by moving a joystick and coordinates of the joystick movement (in X and Y axes) will be recorded at 120 Hz. The IDE, RMSE, movement time, reaction time, travel distance, and the rate learning variables can be then calculated from these X and Y coordinates for each trial. The brain activity change during the rVMA will be described by the relative change of the oxyhemoglobin and deoxyhemoglobin measures done by the fNIRS at 50 Hz. The other variables will be used statistically to describe the groups (specific objective O1.1). In the next table, the reviewer can see all the variables, their units, and the data level of measurement. (Please see also table included in the document 'Response to reviewers'). 

Answer to the choice of statistical test (parametric vs. non parametric)

We agreed with the reviewer that no exploration for normal distribution will be need for categorical and ordinal data (i.e., sex at birth, handedness, PAQ-C score, M-ABC2 score, Conners test score, and TONI-4 score) and specific non-parametric test has to be selected beforehand. Crosstab statistics will be use to explore the differences between groups of sex and handedness (categorical variables), while Kruskal-Wallis tests will be conducted to identify group differences in PAQ-C, M-ABC2, Conners test, and TONI-4 scores (ordinal variables). We clarify the use of these tests in the new version of the manuscript (page 23-24, lines 576-580 and 586-587).

Description in ‘Data analysis’ section is confusing in the sense that ‘how many groups data are dealt here’? If two groups, then why do you need ANOVA? How the measures are to be explored through one-way analysis of variance (ANOVA)? [refer to lines 559-561: Age, sex at birth, BMI, handedness, physical activity engagement, and fitness level (estimated VO2max) measures will be explored through one-way analysis of variance (ANOVA)]. What do you mean by exploring measures through ANOVA? Refer to lines 576-78 where stated that “Differences between the children groups (O1.2 and O1.3) on the motor learning and on the cortical activity (O2.1 and O2.2) through all sets will be explored using a General Lineal Mixed Model (GLMM) with repeated measures”. Are differences between the children groups explored using a General Lineal Mixed Model (GLMM) with repeated measures? How many times repeated observations you have? Please clarify all the aspects of ‘DESIGN’ of the study, first [very clearly]. Remember/mind you that this is a scientific/academic document and so all details should be clearly/correctly communicated (do not take readers’ for granted). Kindly check for the ‘English’ language. Agreed that English is not our mother tongue (definitely not mine, may or may not be yours but certainly not of many readers).

Answer to the comment on the data analysis with respect to the number of groups and the type of test used (i.e., ANOVA vs. t-tests)

In our study protocol, we include two cohorts of children (children with DCD and children TD) that are assigned using partial-within cohort randomization to an exercise group (EX, children who will perform exercise before the motor learning task) or a control group (CON, children who will not perform exercise before the motor learning task). Therefore, we have in total four groups: CON-DCD, CON-TD, EX-DCD, and EX-TD. Please, see the ‘Study design and setting’ section (page 13 lines 306-309), the ‘Participants’ section (page 15, lines 355-360), the ‘Statistical size calculation’ section (page 16, lines 375-376), or the ‘Data analysis’ section (page 24, lines 584-587). 

In this case when four groups are defined (2 cohort (DCD, TD) and 2 exercise (EX, CON)), applying multiple t-tests will not be correct and would induce type I errors. Typically for ratio data, two-way ANOVA with appropriate corrections (e.g., Greenhouse-Geisser, Bonferroni) is the adequate statistical analysis to inform about the main effect of each factor (i.e., cohort and exercise) and the interaction of both factors (possible differences between the four independent groups: CON-DCD, CON-TD, EX-DCD, and EX-TD). Alternatively, cross-tabulations (categorical data) and Kruskal-Wallis test (ordinal data) are more appropriate to compare groups with non-parametric measures. 

Answer to the comments with respect to clarity of the design (e.g., clarifying number of observations).

Thank you for pointing some issues detected in the ‘Data analysis’ section. We modified the text to clarify all the detected problems and following the statistical analyses summarized in the next table. (Please see also table included in the document 'Response to reviewers'). 

Statistics in the study protocol are organized similarly to the table, that is, per objectives: 

• The first paragraph presented the statistical test that will be conducted during participants’ allocation to ensure that this allocation of participants does not create initial group differences that could impact on the results of the 

---

## [Editor Report · Decision Letter 2]

1 Apr 2024

Study protocol to examine the effects of acute exercise on motor learning and brain activity in children with developmental coordination disorder (ExLe-Brain-DCD)

PONE-D-23-24057R2

Dear Dr. Busquets,

We thank you for your patience during this review process. And we appreciate your thoughtful comments in responses to the latest round of reviews. We’re pleased to inform you that your manuscript has been judged scientifically suitable for publication and will be formally accepted for publication once it meets all outstanding technical requirements.

Kind regards,

Bradley R. King

Academic Editor

PLOS ONE
---

## [Editor Report · Acceptance letter]

29 Apr 2024

PONE-D-23-24057R2 

PLOS ONE

Dear Dr. Busquets, 

I'm pleased to inform you that your manuscript has been deemed suitable for publication in PLOS ONE. Congratulations! Your manuscript is now being handed over to our production team.

Kind regards, 

on behalf of

Dr. Bradley R. King 

Academic Editor

PLOS ONE